# TESSAR: Geometry-Aware Active Regression via Dynamic Voronoi Tessellation

**Seong Jin Cho**
Korea Institute of Oriental Medicine
ipcng@kiom.re.kr

**Gwangsu Kim**
Jeonbuk National University
s88012@jbnu.ac.kr

**Junghyun Lee, Hee Suk Yoon, Joshua Tian Jin Tee, Chang D. Yoo***
Korea Advanced Institute of Science and Technology
{jh_lee00,hskyoon,joshuateetj,cd_yoo}@kaist.ac.kr

## Abstract

Active learning improves training efficiency by selectively querying the most informative samples for labeling. While it naturally fits classification tasks–where informative samples tend to lie near the decision boundary–its application to regression is less straightforward, as information is distributed across the entire dataset. Distance-based sampling is commonly used to promote diversity but tends to overemphasize peripheral regions while neglecting dense, informative interior regions. To address this, we propose a Voronoi-based active learning framework that leverages geometric structure for sample selection. Central to our method is the Voronoi-based Least Disagree Metric (VLDM), which estimates a sample's proximity to Voronoi faces by measuring how often its cell assignment changes under perturbations of the labeled sites. We further incorporate a distance-based term to capture the periphery and a Voronoi-derived density score to reflect data representativity. The resulting algorithm, *TESSAR* (TESsellation-based Sampling for Active Regression), unifies interior coverage, peripheral exploration, and representativity into a single acquisition score. Experiments on various benchmarks demonstrate that TESSAR consistently achieves competitive or superior performance compared to prior state-of-the-art baselines.

## 1 Introduction

Active learning aims to improve model performance while reducing labeling costs by selectively querying the most informative data points (Cohn et al., 1996). This is particularly valuable in domains where labeling is expensive or time-consuming. Most active learning research has focused on classification tasks, where various strategies–such as uncertainty sampling (Lewis & Gale, 1994; Balcan et al., 2007), expected error reduction (Yoo & Kweon, 2019), expected model change (Freytag et al., 2014), query-by-committee (Beluch et al., 2018), and Bayesian active learning (Pinsler et al., 2019)–have shown success. A common theme in uncertainty-based methods is to select samples where model predictions are most uncertain. For classification tasks, this often leads to the prioritization of samples near the decision boundary, where uncertainty is typically highest (Kremer et al., 2014; Ducoffe & Precioso, 2018; Cho et al., 2024).

In regression, however, this boundary-centric notion does not apply as all labeled samples contribute to the model globally rather than through local decisions. Consequently, the notions of uncertainty and informativeness must be redefined. Instead of focusing on boundary proximity, informative samples in regression are those that best improve generalization across the entire input space (Wu et al., 2019; Cardenas et al., 2023; Hübotter et al., 2024). Such samples are typically diverse and representative of the data distribution. Thus, existing methods often address this by selecting samples that are far from labeled points (Wu et al., 2019; Ash et al., 2020). This distance-based strategy encourages broad coverage and promotes diversity, but it often oversamples the *periphery*, while

---

*Corresponding author

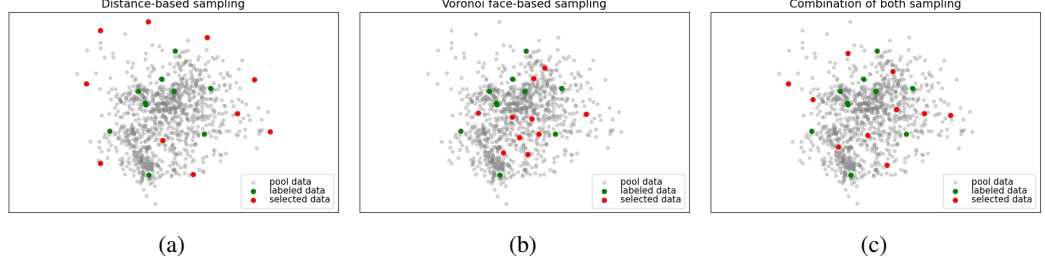

Figure 1: Examples of selected samples by distance-based sampling (a), Voronoi face-based sampling (b), and their combination (c). The combination of both methods effectively captures both the internal and external structure of the data distribution.

overlooking dense and informative *interior regions* (illustrated in Figure 1a). While some methods introduce density-aware corrections (Wu, 2019; Holzmüller et al., 2023), they still offer limited control over interior exploration.

To address this limitation, we consider *Voronoi tessellation*, which partitions the input space into cells around each labeled point (Voronoi, 1908). In the context of Gaussian Process regression, Voronoi tessellation has been used to model discontinuous or heterogeneous geospatial data (Kim et al., 2005; Luo et al., 2021; Pope et al., 2021). Beyond its use in modeling, we propose that Voronoi tessellation serves as an effective surrogate for disagreement-based active classification (Seung et al., 1992; Hanneke, 2014; Cho et al., 2024) in regression. The key intuition is that samples near the boundaries between adjacent cells—known as *Voronoi faces*—often lie in interior regions where the influence of multiple labeled points intersects and competes. Such samples are valuable for enhancing sampling diversity in interior regions, as illustrated in Figure 1b. We further provide theoretical support that samples near Voronoi faces tend to exhibit high prediction variance, indicating greater model uncertainty and, thus, higher potential informativeness.

To efficiently identify these samples, we propose the *Voronoi-based Least Disagree Metric (VLDM)*, which quantifies how often a sample's Voronoi cell assignment changes under perturbations of the labeled site, inspired by Cho et al. (2024). To ensure full spatial coverage, we combine VLDM with a distance-based sampling strategy, as illustrated in Figure 1c. Finally, to complete the triad of effective active learning in regression–informativeness, diversity, and representativity (Wu et al., 2019)–we incorporate a density-based weight derived from Voronoi cell geometry (Holzmüller et al., 2023). Based on these insights, we introduce *TESSAR* (TESsellation-based Sampling for Active Regression), a novel active learning algorithm for regression that combines VLDM, diversity, and representativity into a unified sampling strategy.

**In detail, this paper makes the following key contributions:**

- We introduce the use of Voronoi tessellation for active learning in regression, specifically to target informative samples from interior regions of the input space. We theoretically show that points near Voronoi faces—the boundaries between Voronoi cells—exhibit high prediction variance, making them particularly valuable for improving model performance.

- To *efficiently* identify samples near Voronoi faces, we propose the **Voronoi-based Least Disagree Metric (VLDM)**, a geometric uncertainty measure that quantifies how often a sample's Voronoi cell assignment changes under perturbations of the labeled site.

- We develop **TESsellation-based Sampling for Active Regression (TESSAR)**, a practical active learning algorithm that combines VLDM with strategies for promoting spatial diversity and representativity, balancing exploration of both interior and peripheral regions.

- Extensive experiments across various benchmarks demonstrate that TESSAR achieves competitive or superior performance compared to prior state-of-the-art baselines.

## 2 THE VORONOI-BASED LEAST DISAGREE METRIC (VLDM) FOR INFORMATIVE INTERIOR REGION SAMPLING

As discussed in Figure 1, a key limitation of previous distance-based sampling methods is their tendency to undersample interior regions where the influence of neighboring labeled samples competes. To effectively probe these often-neglected regions, we leverage **Voronoi tessellations** (illustrated in Figure 2). A Voronoi tessellation partitions the input space into distinct cells, where each cell encompasses all points closest to a particular labeled sample (the *site*). Points situated on the boundaries between these cells, known as Voronoi *faces*, are equidistant from two or more sites. Querying points near these faces allows model refinement precisely where the influence of multiple labeled samples converges, enhancing sampling diversity within these interior regions.

### 2.1 PRELIMINARIES AND NOTATION

Let $\mathcal{X}$ and $\mathcal{Y}$ be the feature and label spaces, respectively, with $\mathcal{X} \times \mathcal{Y} \subseteq \mathbb{R}^d \times \mathbb{R}$. We consider multiple regression to learn a function $f : \mathcal{X} \to \mathcal{Y}$. We assume that there exists an unknown ground-truth function $f_\star : \mathcal{X} \to \mathcal{Y}$ that governs the true relationship between inputs and outputs. During active learning we maintain a labeled *instances* $\mathcal{S} = \{\tilde{\boldsymbol{x}}_1, \ldots, \tilde{\boldsymbol{x}}_{|\mathcal{S}|}\} \subset \mathcal{X}$, with noisy labels $y_k = f_\star(\tilde{\boldsymbol{x}}_k) + \eta_k$, where $\eta_k$ is a random variable satisfying $\mathbb{E}[\eta_k] = 0$ and $\eta_k \perp \tilde{\boldsymbol{x}}_k$. A useful geometric perspective for analyzing and organizing the labeled data points in $\mathcal{S}$ is through the concept of Voronoi tessellation. In this context, each point $\tilde{\boldsymbol{x}}_k$ in $\mathcal{S}$ is referred to as a site, and these sites collectively induce a partition of the instance space $\mathcal{X}$ into distinct regions as follows:

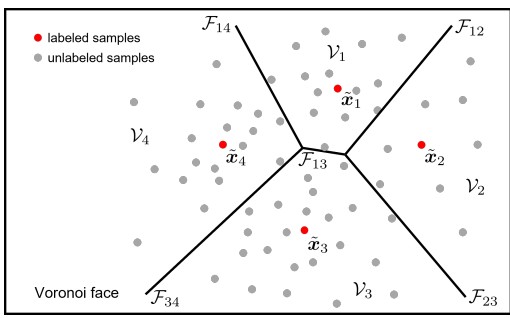

Figure 2: **Illustrative Voronoi tessellation.** Labeled sites (dots) define cells. Points on faces (lines) are equidistant from multiple sites.

$$\mathcal{V}_k := \left\{\boldsymbol{x} \in \mathcal{X} : \|\boldsymbol{x} - \tilde{\boldsymbol{x}}_k\|_2 \leq \|\boldsymbol{x} - \tilde{\boldsymbol{x}}_j\|_2 \text{ for all } j \neq k\right\}, \quad k = 1, \ldots, |\mathcal{S}|. \tag{1}$$

Each convex region $\mathcal{V}_k$ is the *Voronoi cell* of site $\tilde{\boldsymbol{x}}_k$ and contains all points closer to that site than to any other. Whenever two distinct cells $\mathcal{V}_j$ and $\mathcal{V}_k$ meet, their common boundary $\mathcal{F}_{jk} \triangleq \mathcal{V}_j \cap \mathcal{V}_k$ for each $j \neq k$ is called a **Voronoi face**. Points on or near such faces are equidistant to at least two sites, so no single labeled sample dominates their local geometry.

### 2.2 INFORMATIVENESS OF VORONOI FACES: A THEORETICAL PERSPECTIVE

Geometrically, selecting samples near Voronoi faces naturally promotes diversity by focusing on regions between labeled sites. We now argue that these regions are also intrinsically informative from the perspective of model uncertainty.

Formally, suppose both the trained predictor $\widehat{f}$ and the ground-truth function $f_\star$ are $L$-Lipschitz, the observation noise $\eta_k$ is zero-mean, and a "good" event holds with high probability such that $|\widehat{f}(\tilde{\boldsymbol{x}}_k) - f_\star(\tilde{\boldsymbol{x}}_k)| \leq \epsilon$, for some $\epsilon = \epsilon(|\mathcal{S}|) > 0$. This statistical error typically decays as $|\mathcal{S}|^{-\beta}$ for some $\beta > 0$, depending on the noise distribution and function class (Tsybakov, 2009), and can be regarded as small.

For any unlabeled point $\boldsymbol{x}' \in \mathcal{X}$ and labeled site $\tilde{\boldsymbol{x}}_k \in \mathcal{S}$, the triangle inequality gives

$$\left|\widehat{f}(\boldsymbol{x}') - f_\star(\boldsymbol{x}')\right| \leq \left|\widehat{f}(\boldsymbol{x}') - \widehat{f}(\tilde{\boldsymbol{x}}_k)\right| + \left|\widehat{f}(\tilde{\boldsymbol{x}}_k) - f_\star(\tilde{\boldsymbol{x}}_k)\right| + \left|f_\star(\tilde{\boldsymbol{x}}_k) - f_\star(\boldsymbol{x}')\right|$$
$$\leq 2L\|\boldsymbol{x}' - \tilde{\boldsymbol{x}}_k\|_2 + \epsilon,$$

where the second line uses Lipschitzness and the "good" event. Thus,

$$f_\star(\boldsymbol{x}') - \left(2L\|\boldsymbol{x}' - \tilde{\boldsymbol{x}}_k\|_2 + \epsilon\right) \leq \widehat{f}(\boldsymbol{x}') \leq f_\star(\boldsymbol{x}') + \left(2L\|\boldsymbol{x}' - \tilde{\boldsymbol{x}}_k\|_2 + \epsilon\right).$$

Under the "good" event, Popoviciu's inequality (Popoviciu, 1935) then implies

$$\mathrm{Var}[\widehat{f}(\boldsymbol{x}')] \leq \left(2L\|\boldsymbol{x}' - \tilde{\boldsymbol{x}}_k\|_2 + \epsilon\right)^2.$$

As $\epsilon$ does not depend on $k^{1}$, minimizing over $k \in S$ shows that the predictive variance at $\boldsymbol{x}'$ is controlled by the squared distance to the nearest labeled site.

Since predictive variance is governed by the distance to labeled sites, it is natural to sample points that maximize this distance relative to multiple sites—namely, those near Voronoi faces. Under Lipschitzness, each Voronoi cell can be viewed as a region where labels differ only within a bounded range, so points near faces are precisely those where this bounded variation is shared across sites. Such points are unstable: small perturbations of the labeled sites can shift the Voronoi partition and change the site to which they correspond. This instability parallels disagreement-based active classification, where informative samples lie near decision boundaries because small changes in the hypothesis flip their labels (Seung et al., 1992; Hanneke, 2014; Cho et al., 2024). While regression lacks discrete boundaries, Voronoi faces play an analogous role, with samples near them forming natural candidates for informative queries.

## 2.3 VORONOI-BASED LEAST DISAGREE METRIC (VLDM)

Selecting samples near Voronoi faces offers better coverage of the interior regions of the input space, but computing the Voronoi diagram is computationally prohibitive in high-dimensional settings. Specifically, constructing the diagram for $S$ sites in $\mathbb{R}^d$ requires $\mathcal{O}\left(S \log S + S^{\lfloor d/2 \rfloor}\right)$ time (Klee, 1980), which is infeasible for high dimensions. To overcome this challenge, we introduce an efficient surrogate that estimates Voronoi face proximity without computing the diagram.

Let $S \in \mathbb{N}$ denote the number of sites, and let $\mathcal{X}$ be the instance space under consideration. Define $\binom{\mathcal{X}}{S} := \{\mathcal{S} \subset \mathcal{X} : |\mathcal{S}| = S\}$ as the collection of all possible site configurations of size $S$. We fix a feature mapping $\boldsymbol{z} : \boldsymbol{x} \in \mathcal{X} \mapsto \boldsymbol{z}_{\boldsymbol{x}} \in \mathbb{R}^d$, e.g., the final-layer representation from a neural network.

To define Voronoi cells in feature space, we first assume a fixed but arbitrary ordering over the sites within each $\mathcal{S} \in \binom{\mathcal{X}}{S}$. This does not affect the geometry of Voronoi partitions and is used solely to make $h_{\mathcal{S}}$ well-defined. We can then define the Voronoi hypothesis space $\mathcal{H} := \{h_{\mathcal{S}} \mid \mathcal{S} \in \binom{\mathcal{X}}{S}\}$, where we associate each $\mathcal{S} = \{\tilde{\boldsymbol{x}}_i\}_{i \in [S]}$ with a hypothesis over $\mathcal{X}$, $h_{\mathcal{S}} : \mathcal{X} \to [S]$, defined as

$$h_{\mathcal{S}}(\boldsymbol{x}) := \arg\min_{k \in [S]} \left\{ d_{\mathrm{z}}(\boldsymbol{x}, \tilde{\boldsymbol{x}}_k) \triangleq \|\boldsymbol{z}_{\boldsymbol{x}} - \boldsymbol{z}_{\tilde{\boldsymbol{x}}_k}\|_2 \right\}. \tag{2}$$

We refer to $d_{\mathrm{z}}(\cdot, \cdot)$ as the *feature distance*.

**Permutation-invariant alignment.** Since the label assigned by $h_{\mathcal{S}}$ corresponds to the index of the nearest site, the labels themselves are arbitrary up to permutation. For two Voronoi hypotheses $h_{\mathcal{S}}$ and $h_{\mathcal{S}'}$, we define the optimal permutation that attains the maximal overlap between the two Voronoi diagrams as $\pi_{\mathcal{S},\mathcal{S}'} := \arg\min_{\pi \in \mathrm{Sym}(S)} \mathbb{P}_{X \sim \mathcal{D}_{\mathcal{X}}}(h_{\mathcal{S}}(X) \neq \pi \circ h_{\mathcal{S}'}(X))$, where $\mathrm{Sym}(S)$ is the set of permutations over $[S]$. This is analogous to the common practice in clustering evaluation, where accuracy is measured up to label permutations (Lu & Zhou, 2016).

In practice, if $\mathcal{S}'$ is a slight perturbation of $\mathcal{S}$, then $\pi_{\mathcal{S},\mathcal{S}'}$ often corresponds to maintaining the same label indices due to the geometric stability[2] of Voronoi regions under small shifts (Reem, 2011). For instance, for $\mathcal{S} = \{\tilde{\boldsymbol{x}}_k\}_{k \in [S]}$ and small perturbations $\{\boldsymbol{\varepsilon}_i\}_{k \in [S]}$, we would have that $\mathcal{S}' = \{\tilde{\boldsymbol{x}}'_k := \tilde{\boldsymbol{x}}_k + \boldsymbol{\varepsilon}_k\}_{k \in [S]}$, i.e., the site labels do not change from $\mathcal{S}$ to $\mathcal{S}'$ and vice-versa. Thus, from hereon and forth, we will simply drop the dependency on $\pi_{\mathcal{S},\mathcal{S}'}$.

**Voronoi-based Least Disagree Metric.** Inspired by disagreement-based active classification (Hanneke, 2014; Cho et al., 2024), we define the **Voronoi-based Least Disagree Metric (VLDM)** to quantify how easily the Voronoi cell to which a sample belongs changes under slight perturbations of the site configuration.

For each $\mathcal{S} \in \binom{\mathcal{X}}{S}$ and $\boldsymbol{x}_0 \in \mathcal{X}$, the **VLDM** is defined as follows:

$$L(h_{\mathcal{S}}, \boldsymbol{x}_0) := \inf_{h_{\mathcal{S}'} \in \mathcal{H}^{h_{\mathcal{S}}, \boldsymbol{x}_0}} \left\{ \rho(h_{\mathcal{S}'}, h_{\mathcal{S}}) \triangleq \mathbb{P}_{X \sim \mathcal{D}_{\mathcal{X}}}(h_{\mathcal{S}}(X) \neq h_{\mathcal{S}'}(X)) \right\}, \tag{3}$$

---

[1] While instance-dependent errors are possible, we disregard them here for simplicity.

[2] Precisely speaking, a small change of the sites yields a small change in the corresponding Voronoi cells with respect to the Hausdorff distance (Reem, 2011, Theorem 5.1).

---

**Algorithm 1** Empirical Evaluation of VLDM

---

**Input:**
$\boldsymbol{x}$: target sample
$\mathcal{S}^{(0)}$: site configurations
$M$: number of samples for approximation
$\{\sigma_v^2\}_{v=1}^V$, $N$: set of variance and number of perturbation

$L_{\boldsymbol{x}} = 1$
$D_{\boldsymbol{x}}^{(0)} = \min_{\tilde{\boldsymbol{x}} \in \mathcal{S}^{(0)}} d_{\mathrm{z}}(\boldsymbol{x}, \tilde{\boldsymbol{x}})$, $K_{\boldsymbol{x}}^{(0)} = h_{\mathcal{S}^{(0)}}(\boldsymbol{x})$    *(for TESSAR)*
**for** $v = 1$ **to** $V$ **do**
     **for** $n = 1 + (v-1)\lfloor N/V \rfloor$ **to** $v \lfloor N/V \rfloor$ **do**
         Construct $\mathcal{S}^{(n)}$ with $\boldsymbol{z}_{\tilde{\boldsymbol{x}}'} \sim \mathcal{N}(\boldsymbol{z}_{\tilde{\boldsymbol{x}}}, \sigma_v^2 \mathbf{I})$, $\forall \tilde{\boldsymbol{x}} \in \mathcal{S}^{(0)}$
         **if** $h_{\mathcal{S}^{(n)}}(\boldsymbol{x}) \neq h_{\mathcal{S}^{(0)}}(\boldsymbol{x})$ **then**
             $L_{\boldsymbol{x}} \leftarrow \min\{L_{\boldsymbol{x}}, \rho_M(h_{\mathcal{S}^{(n)}}, h_{\mathcal{S}^{(0)}})\}$
         $D_{\boldsymbol{x}}^{(n)} = \min_{\tilde{\boldsymbol{x}} \in \mathcal{S}^{(n)}} d_{\mathrm{z}}(\boldsymbol{x}, \tilde{\boldsymbol{x}})$, $K_{\boldsymbol{x}}^{(n)} = h_{\mathcal{S}^{(n)}}(\boldsymbol{x})$    *(for TESSAR)*
**return:** $L_{\boldsymbol{x}}$ *(for VLDM)*,    $\{D_{\boldsymbol{x}}^{(n)}, K_{\boldsymbol{x}}^{(n)}, \mathcal{S}^{(n)}\}_{n=0}^N$ *(for TESSAR)*

---

where $\mathcal{D}_{\mathcal{X}}$ is the marginal distribution over $\mathcal{X}$ and $\mathcal{H}^{h_{\mathcal{S}}, \boldsymbol{x}_0} := \{h_{\mathcal{S}'} \in \mathcal{H} \mid h_{\mathcal{S}}(\boldsymbol{x}_0) \neq h_{\mathcal{S}'}(\boldsymbol{x}_0)\}$ is the set of Voronoi hypotheses in $\mathcal{H}_N$ that *disagree* with $h_{\mathcal{S}}$ on $\boldsymbol{x}_0$. Again, recall that as long as $\mathcal{S}'$ is obtained from a small perturbation of $\mathcal{S}$, there is no need to explicitly compute $\pi_{\mathcal{S}, \mathcal{S}'}$.

## 2.4   Empirical Evaluation of VLDM

We employ two approximation schemes to compute Eqn. 3 as in Cho et al. (2024). First, we replace $\mathcal{H}^{h_{\mathcal{S}}, \boldsymbol{x}_0}$ with a finite collection of $N$ hypotheses, $\mathcal{H}_N^{h_{\mathcal{S}}, \boldsymbol{x}_0}$. Each $h_{\mathcal{S}'} \in \mathcal{H}_N^{h_{\mathcal{S}}, \boldsymbol{x}_0}$ is generated by perturbing $\mathcal{S}$ using multiple Gaussian noise levels. Specifically, for each variance parameter $\sigma_v^2$ in the predefined set $\{\sigma_v^2\}_{v=1}^V$, we construct perturbed sites $\boldsymbol{z}_{\tilde{\boldsymbol{x}}_i'} \sim \mathcal{N}(\boldsymbol{z}_{\tilde{\boldsymbol{x}}}, \sigma_v^2 \mathbf{I})$ to obtain $\mathcal{S}'$. For each resulting perturbed configuration $\mathcal{S}'$, we include the corresponding hypothesis $h_{\mathcal{S}'}$ in $\mathcal{H}_N^{h_{\mathcal{S}}, \boldsymbol{x}_0}$ whenever $h_{\mathcal{S}'}(\boldsymbol{x}_0) \neq h_{\mathcal{S}}(\boldsymbol{x}_0)$. The use of multiple variances enables VLDM to capture how easily the Voronoi cell to which a sample belongs changes across a range of perturbation magnitudes. Second, we replace $\rho$ with Monte-Carlo approximation with $M$ samples:

$$\rho_M(h_{\mathcal{S}'}, h_{\mathcal{S}}) := \frac{1}{M} \sum_{i=1}^M \mathbb{I}\big[h_{\mathcal{S}'}(X_i) \neq h_{\mathcal{S}}(X_i)\big], \quad X_i \overset{i.i.d.}{\sim} \mathcal{D}_{\mathcal{X}}, \tag{4}$$

where $\mathbb{I}[\cdot]$ is the indicator function. Finally, we define the empirical VLDM as $L_{N,M}(h_{\mathcal{S}}, \boldsymbol{x}_0) := \inf_{h_{\mathcal{S}'} \in \mathcal{H}_N^{h_{\mathcal{S}}, \boldsymbol{x}_0}} \rho_M(h_{\mathcal{S}'}, h_{\mathcal{S}})$. Under certain regularity conditions, its asymptotic consistency is guaranteed (Cho et al., 2024, Theorem 1), which then implies that the ordering of empirically evaluated VLDM values is preserved in probability (Cho et al., 2024, Corollary 1); see Figure 3 in Section 4 for an empirical demonstration of this claim.

Algorithm 1 summarizes the above discussions as a pseudocode for empirically evaluating the VLDM of $\boldsymbol{x}$ for given $\mathcal{S}^{(0)}$. Note that other than the computed empirical VLDM $L_{\boldsymbol{x}}$, the algorithm also outputs other values $\{D_{\boldsymbol{x}}^{(n)}, K_{\boldsymbol{x}}^{(n)}, \mathcal{S}^{(n)}\}_{n=0}^N$; these are used for VLDM-based active learning to be described in Section 3.

## 3   TESSAR: Voronoi Tessellation-based Active Regression

**Acquisition Score.** To enable balanced sampling across both the interior and periphery of the input space, we combine three geometry-based criteria: a VLDM-based weight, a distance score, and a density-aware representativity term. Each component contributes to covering different spatial regions or properties of the input space. We first define a VLDM-based weight:

$$\gamma_{\boldsymbol{x}} = \frac{e^{-\eta_{\boldsymbol{x}}}}{\sum_{\boldsymbol{x}_j \in \mathcal{P}} e^{-\eta_{\boldsymbol{x}_j}}}, \quad \text{where} \quad \eta_{\boldsymbol{x}} = \frac{(L_{\boldsymbol{x}} - L_q)_+}{L_q}. \tag{5}$$

---

**Algorithm 2** Active Learning with TESSAR

---

**Input:**
$\mathcal{L}_0, \mathcal{U}_0$ : Initial labeled and unlabeled samples
$q$ : query size
$T$ : number of acquisition steps

**for** $t = 0$ **to** $T - 1$ **do**
   Train model using $\mathcal{L}_t$
   Set $\mathcal{P} \subseteq \mathcal{U}_t$ and $\mathcal{S}^{(0)} = \{\boldsymbol{x}_i \mid (\boldsymbol{x}_i, y_i) \in \mathcal{L}_t\}$
   Get $L_{\boldsymbol{x}}, \{D_{\boldsymbol{x}}^{(n)}, K_{\boldsymbol{x}}^{(n)}\}_{n=0}^N$ of $\boldsymbol{x} \in \mathcal{P}$ and $\{\mathcal{S}^{(n)}\}_{n=0}^N$ using Algorithm 1
   $\mathcal{Q}_0 \leftarrow \emptyset$
   **for** $j = 1$ **to** $q$ **do**
      Compute $\gamma_{\boldsymbol{x}}, S_{\boldsymbol{x}}$ using Eqn. 5 and 7
      $\tilde{\boldsymbol{x}}_{\text{new}} = \arg\max_{\boldsymbol{x} \in \mathcal{P}} \gamma_{\boldsymbol{x}} * D_{\boldsymbol{x}}^{(0)} * S_{\boldsymbol{x}}$
      $\mathcal{Q}_j \leftarrow \mathcal{Q}_{j-1} \cup \{\tilde{\boldsymbol{x}}_{\text{new}}\}$
      **for** $\boldsymbol{x} \in \mathcal{P}$ **do**
         $L_{\boldsymbol{x}}, \{D_{\boldsymbol{x}}^{(n)}, K_{\boldsymbol{x}}^{(n)}, \mathcal{S}^{(n)}\}_{n=0}^N \leftarrow \textsc{UpdateVLDM}(\boldsymbol{x}, \tilde{\boldsymbol{x}}_{\text{new}}, L_{\boldsymbol{x}}, \{D_{\boldsymbol{x}}^{(n)}, K_{\boldsymbol{x}}^{(n)}, \mathcal{S}^{(n)}\}_{n=0}^N)$
   $\mathcal{L}_{t+1} \leftarrow \mathcal{L}_t \cup \{(\boldsymbol{x}_i, y_i) \mid \boldsymbol{x}_i \in \mathcal{Q}_q\}, \quad \mathcal{U}_{t+1} \leftarrow \mathcal{U}_t \setminus \mathcal{Q}_q$

**subroutine** $\textsc{UpdateVLDM}(\boldsymbol{x}, \tilde{\boldsymbol{x}}_{\text{new}}, L_{\boldsymbol{x}}, \{D_{\boldsymbol{x}}^{(n)}, K_{\boldsymbol{x}}^{(n)}, \mathcal{S}^{(n)}\}_{n=0}^N)$:
   $\mathcal{S}^{(0)} \leftarrow \mathcal{S}^{(0)} \cup \{\tilde{\boldsymbol{x}}_{\text{new}}\}$
   **if** $d_{\text{z}}(\boldsymbol{x}, \tilde{\boldsymbol{x}}_{\text{new}}) < D_{\boldsymbol{x}}^{(0)}$ **then**
      $D_{\boldsymbol{x}}^{(0)} = d_{\text{z}}(\boldsymbol{x}, \tilde{\boldsymbol{x}}_{\text{new}}), \ K_{\boldsymbol{x}}^{(0)} = |\mathcal{S}^{(0)}|$
   **for** $v = 1$ **to** $V$ **do**
      **for** $n = 1 + (v-1)\lfloor N/V \rfloor$ **to** $v\lfloor N/V \rfloor$ **do**
         Sample $\tilde{\boldsymbol{x}}_{\text{new}}^{(n)} \sim \mathcal{N}(\tilde{\boldsymbol{x}}_{\text{new}}, \sigma_v^2 \mathbf{I})$
         $\mathcal{S}^{(n)} \leftarrow \mathcal{S}^{(n)} \cup \{\tilde{\boldsymbol{x}}_{\text{new}}^{(n)}\}$
         **if** $d_{\text{z}}(\boldsymbol{x}, \tilde{\boldsymbol{x}}_{\text{new}}^{(n)}) < D_{\boldsymbol{x}}^{(n)}$ **then**
            $D_{\boldsymbol{x}}^{(n)} = d_{\text{z}}(\boldsymbol{x}, \tilde{\boldsymbol{x}}_{\text{new}}^{(n)}), \ K_{\boldsymbol{x}}^{(n)} = |\mathcal{S}^{(n)}|$
         **if** $K_{\boldsymbol{x}}^{(n)} \neq K_{\boldsymbol{x}}^{(0)}$ **then**
            $L_{\boldsymbol{x}} \leftarrow \min\{L_{\boldsymbol{x}}, \rho_M(h_{\mathcal{S}^{(n)}}, h_{\mathcal{S}^{(0)}})\}$
   **return:** $L_{\boldsymbol{x}}, \{D_{\boldsymbol{x}}^{(n)}, K_{\boldsymbol{x}}^{(n)}, \mathcal{S}^{(n)}\}_{n=0}^N$

---

Here, $L_q$ denotes the $q^{\text{th}}$ smallest VLDM value in the pool data and $(\cdot)_+ = \max\{0, \cdot\}$. This formulation gives exponentially higher weight to samples with smaller VLDM values, encouraging selection near Voronoi faces and thus improving coverage of interior regions. To complement this, we use the sample's shortest feature distance to the sites, referred to as 'DIST':

$$D_{\boldsymbol{x}} = \min_{\tilde{\boldsymbol{x}} \in \mathcal{S}} d_{\text{z}}(\boldsymbol{x}, \tilde{\boldsymbol{x}}), \tag{6}$$

which captures how far the sample is from existing sites. This encourages exploration of under-represented peripheral regions. Together, VLDM and DIST provide coverage across the full input domain and jointly contribute to both informativeness and diversity. However, they may overlook the underlying data distribution. To account for sample density, we incorporate a representativity score inspired by cluster-based sampling (Holzmüller et al., 2023), referred to as 'BIN':

$$S_{\boldsymbol{x}} = \sum_{\boldsymbol{x}' \in \mathcal{P}: h_{\mathcal{S}}(\boldsymbol{x}') = h_{\mathcal{S}}(\boldsymbol{x})} D_{\boldsymbol{x}'}^2. \tag{7}$$

This score is shared across all samples belonging to the same Voronoi cell and reflects the cell's local density based on intra-cell distances. It encourages sampling in more densely populated regions, aligning the query strategy with the overall data distribution. Finally, the three criteria are combined multiplicatively to score each candidate. This unified strategy ensures that selected samples are informative, spatially diverse, and representative of the underlying distribution.

**TESSAR (TESsellation-based Sampling for Active Regression).** We now introduce **TESSAR** (Algorithm 2), a Voronoi tessellation-based active regression algorithm. At each step $t$, the model is trained on $\mathcal{L}_t$ to extract features for the pool data, where the pool is the set of all unlabeled samples,

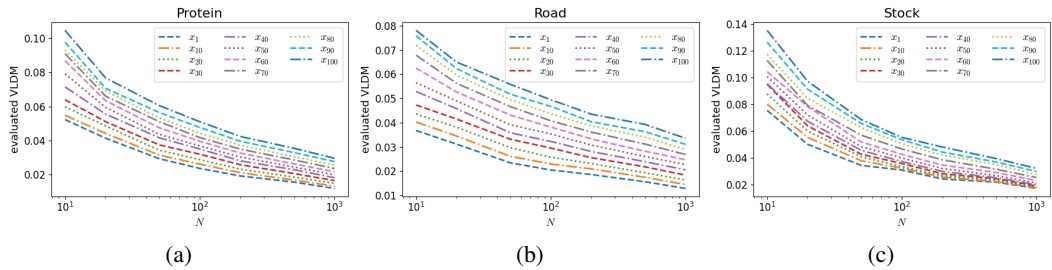

Figure 3: Empirically evaluated VLDMs by Algorithm 1 with respect to the number of perturbations, $N$, on Protein (a), Road (b), and Stock (c) datasets. Observe that the evaluated VLDM monotonically decreases as $N$ increases, and the rank order is well maintained.

$\mathcal{P} \subseteq \mathcal{U}_t$. The labeled inputs serve as sites, $\mathcal{S}^{(0)} = \{\boldsymbol{x}_i \mid (\boldsymbol{x}_i, y_i) \in \mathcal{L}_t\}$. For each $\boldsymbol{x} \in \mathcal{P}$, the quantities $L_{\boldsymbol{x}}, \{D_{\boldsymbol{x}}^{(n)}, K_{\boldsymbol{x}}^{(n)}\}_{n=0}^N$ are computed, and $\{\mathcal{S}^{(n)}\}_{n=0}^N$ is obtained via Algorithm 1. The query set $\mathcal{Q}_0$ is initialized as $\emptyset$. For $j = 1, \ldots, q$, $\gamma_{\boldsymbol{x}}$ and $S_{\boldsymbol{x}}$ are evaluated for each $\boldsymbol{x} \in \mathcal{P}$ using Eqn. 5 and Eqn. 7. The algorithm then selects $\tilde{\boldsymbol{x}}_{\text{new}} \in \mathcal{P}$ maximizing $\gamma_{\boldsymbol{x}} * D_{\boldsymbol{x}}^{(0)} * S_{\boldsymbol{x}}$, which is our acquisition score, and appends $\tilde{\boldsymbol{x}}_{\text{new}}$ to $\mathcal{Q}_j$. Afterwards, $L_{\boldsymbol{x}}, \{D_{\boldsymbol{x}}^{(n)}, K_{\boldsymbol{x}}^{(n)}, \mathcal{S}^{(n)}\}_{n=0}^N$ are updated using subroutine UPDATEVLDM(), described below. Finally, the algorithm queries the labels $y_i$ for all $\boldsymbol{x}_i \in \mathcal{Q}_q$, and proceeds until $t = T - 1$.

Naïvely, one would recompute the VLDM from scratch after every selection within a batch, as each newly selected sample alters the Voronoi structure, causing the Voronoi faces and hence the VLDM values. This would require $N \cdot |\mathcal{P}| \cdot q \cdot (|\mathcal{L}| + (q + 1)/2)$ distance computations. To avoid this inefficiency, TESSAR employs a dynamic update strategy to compute VLDM (subroutine UPDATEVLDM()). It first computes distances between all pool samples and the initial labeled site once, then incrementally updates only the distances involving newly selected samples. This reduces the total number of computations to $N \cdot |\mathcal{P}| \cdot (|\mathcal{L}| + q)$, effectively shaving off a factor of $q$. Nevertheless, runtime still scales with the perturbation budget $N$ and pool size $|\mathcal{P}|$, which may require further optimization in large-scale settings (see Appendix C.1). A simpler margin-based variant–where the margin is defined as the difference between the distances from a sample to its nearest and second-nearest Voronoi centers–avoids this dependence on $N$, but its performance is noticeably weaker than VLDM-based TESSAR (see Appendix C.2. The subroutine UPDATEVLDM() implements this dynamic update: given a newly selected site $\tilde{\boldsymbol{x}}_{\text{new}}$ and $L_{\boldsymbol{x}}, \{D_{\boldsymbol{x}}^{(n)}, K_{\boldsymbol{x}}^{(n)}, \mathcal{S}^{(n)}\}_{n=0}^N$, it computes the distances from $\tilde{\boldsymbol{x}}_{\text{new}}$ (and its perturbations) to the pool data, compares them with existing distances, updates $\{D_{\boldsymbol{x}}^{(n)}, K_{\boldsymbol{x}}^{(n)}\}_{n=0}^N$, and then updates the corresponding $L_{\boldsymbol{x}}$ values. Moreover, a comparison with a static VLDM variant shows that removing these dynamic updates leads to substantially degraded performance, highlighting the necessity of the dynamic strategy (see Appendix C.3).

## 4 EXPERIMENTS

This section presents the empirical evaluation of VLDM and TESSAR. We compare its performance against various baseline algorithms on fourteen tabular datasets. We employ a 2-layer MLP with 512 hidden units. All results represent the average performance over 20 repetitions. Detailed descriptions of datasets and experimental settings are provided in Appendix A.

### 4.1 CONSISTENCY OF VLDM

Figure 3 shows the empirically evaluated VLDMs with respect to $N$ of the Protein, Road, and Stock datasets. We denote $\boldsymbol{x}_i$ as the $i^{\text{th}}$ sample *ordered* by the final evaluated VLDM. The empirically evaluated VLDMs are monotonically decreasing while maintaining rank order as $N$ increases.

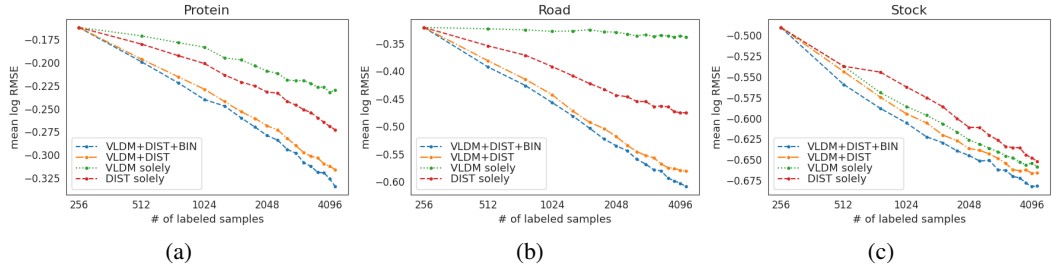

Figure 4: The performance comparison based on the combinations of VLDM ($\gamma_{\boldsymbol{x}}$), DIST ($D_{\boldsymbol{x}}$), and BIN ($S_{\boldsymbol{x}}$) on Protein (a), Road (b), and Stock (c) datasets. The combination of three criteria achieves the best performance.

## 4.2 Effect of the Components in TESSAR

We conduct a comprehensive performance comparison using various combinations of the three criteria. Figure 4 shows the mean log of RMSE with respect to the number of labeled samples on the Protein, Road, and Stock datasets. Individually, VLDM and DIST yield limited performance gains, as each covers only a subset of the input space. However, combining them leads to a significant performance improvement by jointly covering both the interior (via VLDM) and the periphery (via DIST), highlighting their complementary nature. Adding BIN to this combination provides an additional improvement. The inclusion of BIN enhances the sampling strategy by aligning it with the underlying data distribution, particularly in dense regions. Based on these results, we adopt a unified selection strategy that iteratively selects the sample that maximizes the product of the three scores.

## 4.3 Comparing TESSAR to Baseline Algorithms

We now compare the performance of the proposed TESSAR with various baselines.

**Baseline algorithms** Each baseline algorithm is denoted as follows: 'Rand': random sampling, 'Coreset': core-set selection (Sener & Savarese, 2018), 'ProbCov': maximizing probability coverage (Yehuda et al., 2022), 'BALD': Bayesian active learning by disagreement (Houlsby et al., 2011), 'BatchBALD': mutual information between a joint of multiple data points and the model parameters (Kirsch et al., 2019), 'BADGE': batch active learning by diverse gradient embeddings Ash et al. (2020), 'BAIT': batch active learning via information metrics (Ash et al., 2021), 'ACS-FW': active Bayesian coresets with Frank-Wolfe optimization (Pinsler et al., 2019), and 'LCMD': largest cluster maximum distance (Holzmüller et al., 2023).

**Performance comparison *across* datasets** The performance profile (Dolan & Moré, 2002) and penalty matrix (Ash et al., 2020) are utilized for comprehensive comparisons across all datasets. The details of the performance profile and penalty matrix are described in Appendix B. Figure 5a shows the performance profile for regression of all algorithms with respect to $\delta$. TESSAR consistently maintains the highest $R_A(\delta)$ across all considered $\delta$ values. Notably, $R_{\text{TESSAR}}(0) = 41\%$, significantly exceeding the values of other algorithms, including LCMD (29%). Figure 5b further supports TESSAR's superiority. In the first row, TESSAR outperforms all the other algorithms, including LCMD (1.0). Similarly, the first column shows that most algorithms fail to outperform TESSAR, with a maximum penalty of 0.3.

**Performance comparison *per* dataset** Table 1 presents the mean of performance differences (RMSE relative to Random) averaged over repetitions and steps. The negative values indicate better performance than Random. We observe that TESSAR consistently performs best or is comparable to other algorithms across all datasets.

## 5 Related Works

In active learning for regression, a variety of sampling strategies have been developed, each balancing informativeness, diversity, and representativeness in different ways. *Query by committee*

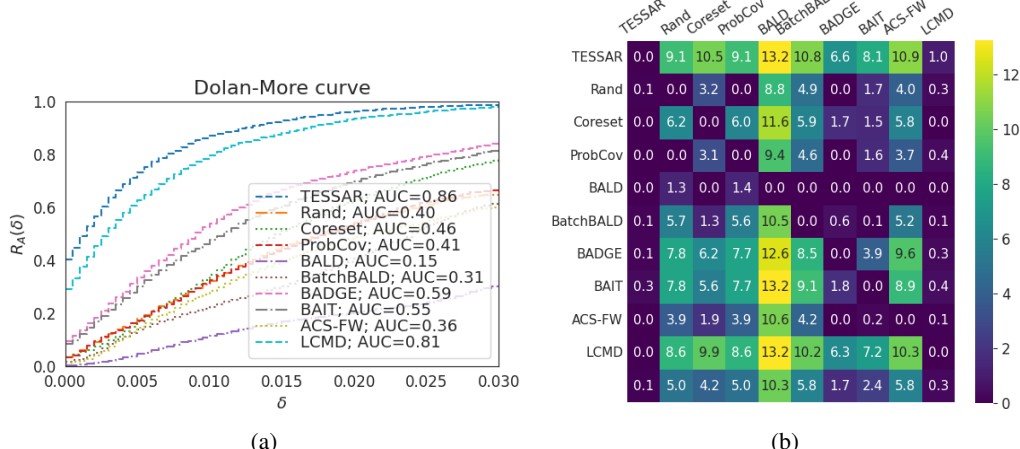

(a)                                           (b)

Figure 5: The performance comparison across datasets. (a) The performance profile results. The AUC value is expressed as a percentage. TESSAR consistently maintains the highest performance across all considered $\delta$ values. (b) The penalty matrix results.

Table 1: The mean of the repetition-wise averaged performance (RMSE) differences, relative to Random, over the entire steps. The negative value indicates higher performance than Random. (**bold+underlined**: best performance, **bold**: second-best performance)

|  | TESSAR | Coreset | ProbCov | BALD | BatchBALD | BADGE | BAIT | ACS-FW | LCMD |
|---|---|---|---|---|---|---|---|---|---|
| CT slices | **-0.0679** | -0.0504 | -0.0040 | 0.0227 | -0.0435 | -0.0395 | -0.0565 | 0.0028 | **-0.0679** |
| Diamonds | **-0.0110** | -0.0094 | 0.0007 | -0.0019 | -0.0079 | -0.0091 | **-0.0106** | -0.0033 | -0.0105 |
| Friedman | -0.0030 | **-0.0033** | 0.0001 | 0.0031 | -0.0026 | -0.0011 | **-0.0049** | -0.0017 | -0.0026 |
| KEGG undir | **-0.1615** | -0.1435 | 0.0088 | 0.0197 | -0.0960 | -0.1297 | -0.1214 | -0.0668 | **-0.1598** |
| Methane | **-0.0520** | -0.0346 | -0.0274 | 0.0251 | -0.0185 | -0.0432 | -0.0355 | -0.0265 | **-0.0510** |
| MLR kNN | **-0.1201** | -0.0192 | 0.0031 | 0.0906 | -0.0580 | -0.0925 | -0.0872 | 0.0006 | **-0.1182** |
| Online video | **-0.1183** | -0.1000 | -0.0001 | -0.0655 | -0.0943 | -0.0991 | -0.0946 | -0.0718 | **-0.1130** |
| Protein | **-0.0082** | 0.0110 | -0.0003 | 0.0308 | 0.0207 | **-0.0075** | 0.0016 | 0.0065 | -0.0054 |
| Query | **-0.0047** | 0.0004 | -0.0001 | 0.0262 | 0.0121 | -0.0023 | -0.0029 | 0.0023 | **-0.0054** |
| Road | **0.0001** | 0.0176 | 0.0000 | 0.1337 | 0.0441 | **-0.0004** | 0.0109 | 0.0294 | 0.0036 |
| SARCOS | **-0.0215** | -0.0080 | -0.0004 | 0.0139 | -0.0016 | -0.0133 | -0.0149 | -0.0093 | **-0.0188** |
| SGEMM | **-0.0141** | -0.0004 | -0.0015 | 0.0817 | 0.0114 | -0.0045 | -0.0038 | 0.0129 | **-0.0121** |
| Stock | **-0.0051** | 0.0046 | -0.0003 | 0.0100 | 0.0075 | **-0.0036** | -0.0001 | 0.0003 | -0.0028 |
| WEC Sydney | **-0.0001** | 0.0163 | 0.0001 | 0.0229 | 0.0045 | **-0.0009** | 0.0004 | 0.0021 | 0.0010 |

*(QBC)* methods (Burbidge et al., 2007; Fazakis et al., 2020) select samples with the highest predictive disagreement among an ensemble of regressors. While they effectively capture uncertainty in well-specified models, they suffer in the presence of noise or model misspecification and incur high computational cost due to repeated model training. *Model change*-based algorithms (Cai et al., 2017; Park & Kim, 2020) select samples that are expected to induce the greatest change in model parameters, often approximated via gradient information. These methods directly target model improvement but typically require ensemble estimation and suffer from high computational complexity, particularly in batch settings. *Black box* approaches (Kirsch, 2023) estimate predictive uncertainty using covariance kernels derived from ensemble predictions. They are compatible with non-differentiable models and achieve strong empirical performance, but rely on ensemble diversity, which may degrade in low-variance models such as random forests or boosted ensembles. *Inverse distance*-based methods (Bemporad, 2023) combine model uncertainty and spatial exploration using inverse-distance weighting. These methods avoid model retraining and generalize to both pool- and population-based settings, but their reliance on distance heuristics limits their effectiveness in high-dimensional or discontinuous spaces. *Greed sampling* methods (Wu et al., 2019) select samples that maximize diversity in the input or output space. These methods are model-aware

and promote label spread and representativity, but require model updates after each selection, increasing computational overhead. *Distribution/coverage*-based methods (Sener & Savarese, 2018; Yehuda et al., 2022; Bae et al., 2024) formulate selection as covering the unlabeled distribution in a learned feature space. Both are label-agnostic and transfer to regression by measuring distances in task-relevant embeddings. However, performance hinges on the quality of the embedding/metric and the choice of coverage radius, and the construction of the graph or mixed integer program (MIP) can be costly for very large pools. *Clustering*-based methods (Wu, 2019; Holzmüller et al., 2023) select samples based on coverage and diversity in feature space. Wu (2019) proposes a sequential representative-diverse (RD) framework using $k$-means clustering, optionally combined with uncertainty measures. Holzmüller et al. (2023) introduce LCMD, a modular batch-mode method that leverages neural tangent kernels and clustering to balance core sampling principles. While effective, both methods may suffer from high runtime due to repeated clustering or kernel computations. TESSAR shares LCMD's use of diversity- and density-oriented components, but fundamentally differs in its VLDM term, which targets samples near Voronoi faces rather than cluster peripheries. This geometric mechanism prevents selection of outer-boundary points–where no Voronoi faces form–and instead reliably guides sampling toward informative interior regions.

More broadly, recent work in adjacent areas highlights a converging theme: efficiency gains come from exploiting *structure* to decide where supervision, optimization, or feedback should be applied. For example, diffusion/generative modeling benefits from timestep- or stepwise-structured optimization and preference signals (Hong et al., 2025; Tee et al., 2025; Pham et al., 2025), and preference optimization can focus on confidence- or preference-critical regions of the signal (Yoon et al., 2025a;b). In multimodal decision-making, sample/feedback efficiency is increasingly achieved by leveraging large vision-language models for distillation or reward construction (Lee et al., 2025; Luu et al., 2025a;b). Finally, structure-aware robustness has been explored in temporal/occlusion or sequential domains (Eom et al., 2025; Gao et al., 2025; Kang et al., 2024; Yoon et al., 2025c), reinforcing the general idea that leveraging geometry/structure can systematically improve data efficiency. These trends support our approach in active regression: using geometric structure induced by the Voronoi faces to identify interior regions where newly queried labels are most likely to yield large gains.

## 6 CONCLUSION

This paper proposes TESSAR, a geometry-driven active learning framework for regression based on Voronoi tessellation. At the core of TESSAR is the Voronoi-based Least Disagree Metric (VLDM), which captures interior structure by identifying samples that lie near Voronoi faces. To ensure full spatial coverage, TESSAR combines VLDM with a distance-based score that promotes exploration of peripheral regions and a region-level representativity term that reflects local data density. This unified strategy enables the model to acquire informative samples from both the interior and periphery while aligning with the overall data distribution. While classification-based methods like LDM-S rely on label-defined decision boundaries and are inherently limited to supervised settings, TESSAR selects samples purely based on geometric relationships among input instances. Empirical evaluations across multiple regression benchmarks show that TESSAR consistently achieves competitive or superior performance.

We conclude by highlighting several promising directions for future work. Although dynamically updating the Voronoi structure in VLDM improves the efficiency of TESSAR, it remains computationally costly since each new sample perturbs the structure even within the same batch. Developing a more scalable variant of TESSAR, or even a new algorithmic principle inspired by our Voronoi-style intuition, is an important next step. Beyond active learning, one natural extension is to investigate how Voronoi-based sample acquisition can guide pseudo-labeling or cluster selection in semi-supervised and unsupervised learning, where geometry rather than label information plays a central role. Finally, while we implicitly assume homoskedasticity, it remains an intriguing open question whether similar Voronoi-style principles can be extended to heteroskedastic settings.

### ACKNOWLEDGMENTS

We thank the anonymous reviewers for their helpful comments. This work was partly supported by the Institute of Information & communications Technology Planning & Evaluation (IITP) grant

funded by the Ministry of Science and ICT (MSIT) (No. RS-2022-II220184, RS-2019-II190075) and partly supported by Global Learning & Academic research institution for Master's·PhD students, and Postdocs (LAMP) Program of the National Research Foundation of Korea (NRF) grant funded by the Ministry of Education (MOE) (No. RS-2024-00443714). This work was partly supported by grant number (No. KSN2512011) from the Korea Institute of Oriental Medicine (KIOM), provided by the Commercialization Promotion Agency for R&D Outcomes (COMPA) grant funded by the MSIT (No. RS-2025-17632971), conducted at the KIOM (No. NSN2521040).

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

APPENDIX

# A EXPERIMENTAL SETTINGS

Table 2: Overview of used datasets.

| Short Name | Citation | Source | Training size | Test size | # of features |
|---|---|---|---|---|---|
| CT slices | (Graf et al., 2011) | UCI | 42,800 | 10,700 | 379 |
| Diamonds | | OpenML | 43,152 | 10,788 | 29 |
| Friedman | (Friedman, 1991) | OpenML | 32,615 | 8,153 | 10 |
| KEGG under | (Shannon et al., 2003) | UCI | 51,687 | 12,921 | 27 |
| Methane | (Ślęzak et al., 2018) | OpenML | 200,000 | 300,000 | 33 |
| MLR kNN | | OpenML | 89,403 | 22,350 | 132 |
| Online video | (Deneke et al., 2014) | UCI | 55,028 | 13,756 | 26 |
| Protein | | OpenML | 36,584 | 9,146 | 9 |
| Query | (Anagnostopoulos et al., 2018) | UCI | 160,000 | 40,000 | 4 |
| Road | (Kaul et al., 2013) | UCI | 200,000 | 234,874 | 2 |
| SARCOS | (Vijayakumar & Schaal, 2000) | GPML | 35,588 | 8,896 | 21 |
| SGEMM | (Ballester-Ripoll et al., 2019) | UCI | 193,280 | 48,320 | 14 |
| Stock | | OpenML | 47,240 | 11,809 | 9 |
| WEC Sydney | (Neshat et al., 2018) | UCI | 57,600 | 14,400 | 48 |

The datasets, deep network architecture, and experimental setup follow the framework proposed by Holzmüller et al. (2023). Table 2 presents an overview of datasets used. We selected 14 tabular regression datasets from different sources. A fully connected neural network with two hidden layers, each comprising 512 neurons ($L = 3, d_1 = d_2 = 512$), is employed for all experiments. The neural tangent parameterization is used in conjunction with the ReLU activation function. All biases are initialized to zero, and weights are independently sampled from $\mathcal{N}(0, 1)$. Model training is performed using the Adam optimizer with default hyperparameters $\beta_1 = 0.9, \beta_2 = 0.999$. The initial learning rate is set to $0.375$ and is decayed linearly to zero throughout training. A batch size of 256 and a total of 256 training epochs are used. After each epoch, the RMSE is evaluated on a validation set of $1,024$ samples. For all datasets, the number of initial labeled samples is set to 256. At each active learning step, 256 samples are queried, and the process is repeated for 16 steps, resulting in a final labeled set of $4,352$ samples. In TESSAR, the number of perturbations $N$ is set to 100, and $\sigma$ is increased in the order of $\{0.0002, 0.0004, 0.0006, 0.0008, 0.001, 0.002, 0.004, 0.006, 0.008, 0.01\}$, with every 10 perturbations. All experiments are conducted on NVIDIA TITAN Xp GPUs with 12GB of memory. We use PyTorch 3.7 with CUDA 10.0.

# B PERFORMANCE PROFILE AND PENALTY MATRIX

## B.1 PERFORMANCE PROFILE

Let $\mathrm{err}_A^{D,r,t}$ denote the RMSE of alrogirhm $A$ at step $t \in T_D$, for dataset $D$ and repetition $r \in [R]$, and define the performance gap as $\Delta_A^{D,r,t} = \mathrm{err}_A^{D,r,t} - \min_{A'}(\mathrm{err}_{A'}^{D,r,t})$. Here, $T_D$ is the number of steps for dataset $D$, and $R$ is the total number of repetitions. Then, the performance profile is defined as:

$$R_A(\delta) := \frac{1}{n_D} \sum_D \left[ \frac{\sum_{r,t} \mathbb{I}(\Delta_A^{D,r,t} \leq \delta)}{RT_D} \right],$$

where $n_D$ is the number of datasets. Intuitively, $R_A(\delta)$ is the fraction of cases where the performance gap between algorithm $A$ and the best competitor is less than $\delta$. Specifically, when $\delta = 0$, $R_A(0)$ is the fraction of cases in which algorithm $A$ performs the best.

## B.2 PENALTY MATRIX

For each dataset, step, and each pair of algorithms $(A_i, A_j)$, we collect $R$ RMSE values $\{\text{err}_i^r\}_{r=1}^R$ and $\{\text{err}_j^r\}_{r=1}^R$ respectively. We compute the $t$-score as $t = \sqrt{R}\bar{\mu}/\bar{\sigma}$ where $\bar{\mu} = \frac{1}{R}\sum_{r=1}^R(\text{err}_i^r - \text{err}_j^r)$ and $\bar{\sigma} = \sqrt{\frac{1}{R-1}\sum_{r=1}^R(\text{err}_i^r - \text{err}_j^r - \bar{\mu})^2}$. In this framework, $A_i$ is said to beat $A_j$ if $t < -2.776$, and vice versa if $t > 2.776$. When $A_i$ beats $A_j$, a penalty of $1/T_D$ is accumulated to $P_{i,j}$, and similarly for the reverse case. Summing the penalties across datasets yields the final penalty matrix.

# C ADDITIONAL RESULTS

## C.1 RUNTIME COMPARISON *per* DATASET

Table 3: The mean of runtime (sec) for each algorithm and each dataset.

|  | TESSAR | Coreset | ProbCov | BALD | BatchBALD | BADGE | BAIT | ACS-FW | LCMD |
|---|---|---|---|---|---|---|---|---|---|
| CT slices | 235.8 | 166.1 | 3,412.1 | 188.0 | 174.3 | 151.4 | 179.7 | 175.2 | 168.7 |
| Diamonds | 255.1 | 205.8 | 4,033.6 | 169.3 | 207.1 | 203.3 | 273.9 | 202.1 | 207.0 |
| Friedman | 287.7 | 202.7 | 4.398.7 | 169.9 | 207.4 | 201.3 | 264.1 | 200.0 | 270.6 |
| KEGG undir | 247.2 | 212.1 | 4,297.9 | 172.5 | 215.5 | 169.2 | 286.9 | 209.2 | 273.9 |
| Methane | 482.9 | 232.9 | 4,518.1 | 171.3 | 190.4 | 218.0 | 378.6 | 211.5 | 245.3 |
| MLR kNN | 367.3 | 213.4 | 4,424.7 | 167.7 | 216.5 | 198.3 | 306.5 | 197.7 | 288.3 |
| Online video | 254.7 | 214.9 | 4,375.2 | 170.3 | 216.8 | 169.5 | 295.9 | 209.6 | 279.7 |
| Protein | 286.1 | 206.7 | 4,373.7 | 170.4 | 208.3 | 205.1 | 279.4 | 204.9 | 275.5 |
| Query | 423.0 | 237.4 | 4.396.2 | 175.8 | 231.8 | 196.1 | 344.0 | 204.0 | 264.2 |
| Road | 547.8 | 236.5 | 3.676.3 | 174.5 | 214.1 | 194.7 | 378.4 | 213.4 | 229.0 |
| SARCOS | 249.9 | 207.2 | 4.340.1 | 168.2 | 208.8 | 204.1 | 276.9 | 204.1 | 276.8 |
| SGEMM | 487.8 | 234.5 | 3,873.1 | 179.7 | 226.9 | 182.5 | 373.0 | 206.8 | 217.4 |
| Stock | 313.1 | 211.9 | 4.347.7 | 172.4 | 212.4 | 207.1 | 285.2 | 207.4 | 212.4 |
| WEC Sydney | 269.6 | 207.9 | 4.237.3 | 174.5 | 204.4 | 194.3 | 286.7 | 191.3 | 278.6 |

Table 3 presents the mean runtime (in seconds) for each algorithm across datasets. For TESSAR, the runtime remains comparable to other strong-performing algorithms such as BADGE, BAIT, and LCMD on smaller datasets. However, as the dataset size increases, TESSAR's runtime grows more noticeably. Nonetheless, the difference typically remains within a few minutes and is negligible relative to the overall time required for labeling. This additional computational cost arises from TESSAR's deliberate design to explore regions–particularly interior areas–that are often underrepresented by conventional sampling methods. Rather than a drawback, this represents a worthwhile trade-off, as TESSAR consistently achieves superior performance in exchange for modest increases in computation. Our results further demonstrate that sampling from interior regions yields meaningful gains in regression tasks, underscoring the importance of this geometric perspective. We hope this finding motivates future research toward more computationally efficient algorithms that retain the benefits of interior-region-aware sampling.

## C.2 VLDM-BASED VS MARGIN-BASED TESSAR

To assess whether TESSAR's performance gains derive from the VLDM formulation or could be matched by simpler geometric uncertainty measures, we replaced VLDM with a margin-based proxy. The margin is defined as the difference between the distances from a sample to its nearest and second-nearest Voronoi centers–an inexpensive approximation of local geometric ambiguity. We then ran TESSAR using this margin score in place of VLDM. Figure 6 shows that margin-based TESSAR incurs notable performance degradation across datasets, despite its lower computational cost. These results demonstrate that the improvements achieved by TESSAR cannot be reproduced by a simple geometric heuristic, thereby validating the necessity of VLDM's design.

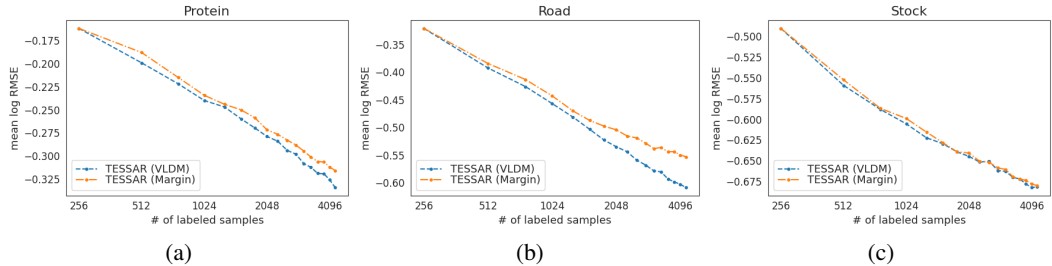

(a)          (b)          (c)

Figure 6: Comparison of VLDM-based and margin-based TESSAR on Protein (a), Road (b), and Stock (c) datasets. The margin is defined as the difference between the distances from a sample to its nearest and second-nearest Voronoi centers. TESSAR using VLDM consistently outperforms its margin-based variant.

### C.3 EFFECT OF DYNAMIC VS. STATIC VORONOI UPDATES

To assess the impact of dynamic tessellation in TESSAR, we included static VLDM and static TES-SAR baselines, in which Voronoi centers remain fixed after initial construction. This setting enables a direct comparison between geometry-aware sampling with and without adaptive updates. Figure 7 shows that static VLDM fails to meaningfully reduce error across datasets, indicating that a fixed Voronoi structure is insufficient for guiding informative selection. Static TESSAR performs reasonably well due to its combined scoring components, but dynamic TESSAR consistently achieves lower error on all datasets, demonstrating that updating Voronoi partitions throughout the acquisition process provides a clear performance benefit.

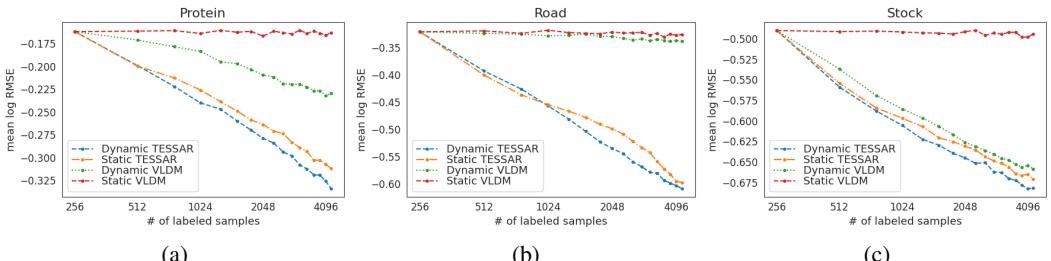

(a)          (b)          (c)

Figure 7: Comparison of dynamic and static VLDM on Protein (a), Road (b), and Stock (c) datasets. While dynamic VLDM shows gradual improvement as more samples are labeled, static VLDM fails to improve across all datasets. Dynamic TESSAR also shows improved performance compared to its static counterpart, validating the importance of adaptive geometry-aware sampling strategies.

### C.4 ROBUSTNESS TO HYPERPARAMETER

In TESSAR, the primary hyperparameter is the number of perturbations, denoted by $N$. This hyperparameter controls the number of perturbed hypotheses sampled during the empirical evaluation of the VLDM, directly affecting both the estimation accuracy of the metric and the overall computational cost. Figure 8 presents the mean log RMSE with respect to $N \in \{10, 50, 100, 500, 1000\}$, and there is no significant performance difference. This suggests that TESSAR is robust to the choice of $N$.

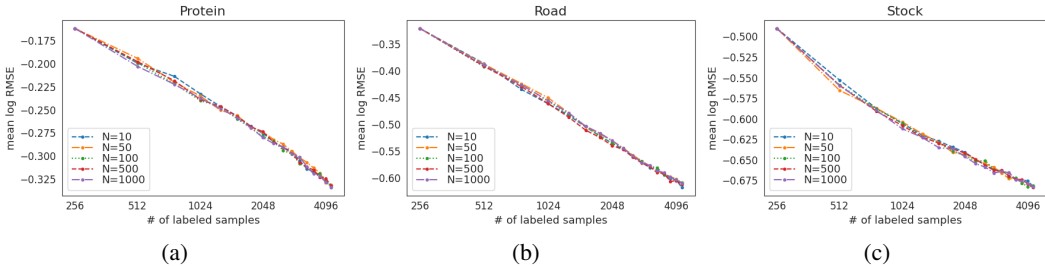

Figure 8: The mean log RMSE with respect to the number of perturbations on Protein (a), Road (b), and Stock (c) datasets. There is no significant performance difference.

## D   THE USE OF LARGE LANGUAGE MODELS (LLMS)

We used LLM to improve grammar and wording. The authors reviewed all edits and take full responsibility for the content.

