# OpenReview forum: "TESSAR: Geometry-Aware Active Regression via Dynamic Voronoi Tessellation"
_ICLR.cc/2026/Conference — ICLR 2026 Poster_

### Official Review · Reviewer_w4R2 · 2025-10-25

**Soundness:** 4
**Presentation:** 3
**Contribution:** 4
**Rating:** 8
**Confidence:** 2

**Summary:**

This paper proposes TESSAR, a geometry-aware active learning framework for regression. The key idea is to model uncertainty through VLDM, which measures the geometric instability of samples under small perturbations of labeled points. By combining VLDM with distance and density terms, the method dynamically selects informative samples in a model-agnostic manner.

**Strengths:**

1. The paper proposes a novel geometric perspective, which is interesting. The use of Voronoi tessellation and the proposed VLDM provide an innovative and well-motivated formulation of uncertainty for regression tasks.
2. The method achieves notable performance gains across various datasets and baselines.

**Weaknesses:**

1. The paper should include a static Voronoi or less frequent update baseline to show the effect of dynamic tessellation. Also what is the selection stragegy of Gaussian perturbation parameter?
2. It would be helpful to include an ablation replacing VLDM with simpler geometric proxies such as nearest-neighbor distance or local label variance. This would clarify whether the performance gains truly stem from the proposed VLDM formulation or can be achieved by simpler uncertainty measures.
2. Voronoi-based geometry can degrade in high-dimensional spaces due to distance concentration. Is the proposed approach still effective in high-dimensional regression tasks?

**Questions:**

See weakness.

---

> ### Author Response · Authors · 2025-11-26
>
> We sincerely thank the reviewer for their careful reading, for highlighting multiple strengths of our paper, and for providing several constructive comments. We are particularly pleased with the positive outlook given to our work. Below, we address each concern point by point and clarify the points raised in the review.
>
> ---
>
> ### W1. Ablation on static vs. dynamic update baselines for tessellation
>
> Thank you for the suggestion. We added static VLDM and static TESSAR baselines to isolate the effect of dynamic tessellation. Static TESSAR performs reasonably well, but dynamic TESSAR consistently achieves better performance across all datasets. Static VLDM, in contrast, shows limited improvement, confirming that adaptive Voronoi updates play a crucial role in TESSAR’s effectiveness. We have added these results in Appendix C.3.
>
> ---
>
> ### W2. Choice of Gaussian perturbations
> In our method, a Gaussian perturbation is applied to the Voronoi centers to generate perturbed hypotheses, which are used to estimate the VLDM. However, choosing an appropriate sigma is non-trivial. If sigma is too small, the perturbed centers remain nearly identical to the original ones, causing most samples to remain in the same Voronoi cell. In such cases, VLDM becomes undefined or uninformative, as there’s no effective variation in region assignments. If sigma is too large, the perturbation cases Voronoi regions to fluctuate excessively, leading to random or overly noisy assignments–again undermining the reliability of disagreement-based selection.
>
> To avoid both extremes without costly hyperparameter tuning, we adopt a multi-scale perturbation strategy as described in Appendix A. A fixed set of sigma values is used to perturb Voronoi centers:
> {0.0002, 0.0004, 0.0006, 0.0008, 0.001, 0.002, 0.004, 0.006, 0.008, 0.01}
> Each sigma value generates a set of perturbed Voronoi centers, and the resulting collection of Voronoi diagrams is used to evaluate VLDM. This ensemble-based approach ensured that the computed disagreement reflects both small-scale local sensitivity and larger structural shifts, enabling robust sample selection without dataset-specific tuning or dynamic adjustment during training. By covering a broad perturbation spectrum, the method balances stability and sensitivity, producing more informative and reliable estimates for active regression.
>
> ### W3. Ablation on replacing VLDM with simpler geometric proxies
> Thank you for this insightful suggestion. We agree that ablations replacing VLDM with simpler proxies are important to highlight the specific contribution of our formulation:
> - **Nearest-Neighbor Distance:** We believe we have addressed the comparison with "nearest-neighbor distance" in Section 4.2. There, we show that the DIST baseline (which relies solely on the shortest feature-to-Voronoi-center distance) is consistently suboptimal compared to DIST+VLDM. This demonstrates that geometric proximity alone is insufficient and must be integrated with an uncertainty measure, namely, our proposed VLDM.
> - **Margin:** Additionally, we have considered an alternate notion of nearest-neighbor distance that we refer to as “margin”. This is defined as the difference between the distances from a sample to its nearest and second-nearest Voronoi centers. As shown in Figure 6 in Appendix C.2, VLDM-based TESSAR outperforms Margin-based TESSAR, demonstrating that the performance gains stem from our proposed VLDM formulation.
>
> ---
>
> ### W4. Effectiveness of TESSAR in high-dimensional regression tasks
> Thank you for this thoughtful question. We agree that distance concentration is a valid concern in high-dimensional spaces. However, our approach remains effective for two key reasons:
> - **Feature Space vs. Input Space:** We apply Voronoi-based geometry on the learned feature space (the last-layer representations), not the raw high-dimensional input space. In our experiments, this feature dimension is $d=512$.
> - **Structure of Learned Features:** Unlike random high-dimensional data, learned neural representations typically lie on lower-dimensional manifolds with meaningful semantic clustering. Consequently, the distance metrics in this space remain informative for defining Voronoi regions, as evidenced by the strong performance in our experiments.
>
> Furthermore, for tasks with extremely high-dimensional inputs, our method is compatible with any architecture that projects data into a moderate-dimensional bottleneck or feature layer before the final regression head.

---

### Official Review · Reviewer_5qY9 · 2025-10-29

**Soundness:** 4
**Presentation:** 4
**Contribution:** 3
**Rating:** 8
**Confidence:** 4

**Summary:**

This paper introduces TESSAR, an active learning framework for regression tasks that leverages Voronoi tessellation to improve sample selection. The core innovation is the Voronoi-based Least Disagree Metric (VLDM), which identifies informative samples near Voronoi faces in the input space, addressing the limitations of traditional distance-based methods that often overlook dense interior regions. VLDM is combined with a distance score for peripheral exploration and a density-based representativity term, resulting in a unified acquisition function. The authors provide theoretical motivation linking Voronoi faces to high predictive variance, along with an efficient approximation for VLDM computation. Empirical evaluations on 14 tabular regression datasets show that TESSAR matches or surpasses state-of-the-art baselines such as LCMD and BADGE in terms of RMSE, although its runtime increases with larger datasets, reducing efficiency, a limitation the authors explicitly address.

**Strengths:**

The use of Voronoi tessellation as a geometric approximation for disagreement-based sampling in regression is a well-motivated idea. Unlike classification tasks with clear decision boundaries, regression lacks such structures, and this paper elegantly adapts the concept via VLDM. The theoretical analysis in Section 2.2, showing that points near Voronoi faces exhibit high variance under Lipschitz assumptions, provides solid grounding.

TESSAR integrates informativeness, diversity, and representativity into a single score, with efficient dynamic updates (Algorithm 2) to avoid recomputing VLDM naively. The empirical consistency of VLDM (Figure 3) and ablation studies (Figure 4) clearly show the complementary benefits of the components in TESSAR.

Evaluations on diverse datasets (e.g., Protein, Road, Stock) using performance profiles and penalty matrices (Figure 5, Table 1) highlight TESSAR's consistent superiority. It achieves the highest RA(0) of 41% in performance profiles, outperforming LCMD (29%). Runtime comparisons (Table 3) indicate it's competitive with baselines, with increases justified by better performance.

The paper is well-written, with clear pseudocode, evaluations and detailed appendices on datasets and metrics, and thoughtful discussion of limitations (e.g., computational cost in large pools, homoskedasticity assumption).

**Weaknesses:**

TESSAR's runtime scales with pool size and perturbations, making it slower on very large datasets (e.g., 547s on Road vs. ~150s for Coreset). The authors acknowledge this and suggest optimizations, but more scaling experiments (e.g., on million-scale data) could strengthen the case. Active Learning is designed for large data sets.

The related works section is comprehensive but could better highlight differences from clustering-based methods like LCMD.

**Questions:**

Can we pre-evaluate how much TESSAR will outperform random sampling on a given dataset?

 Following LCMD’s analysis (Holzmüller et al., 2023), which showed that the ratio of initial RMSE to MAE on a small training set strongly predicts the benefit of LCMD-TP over random selection, a similar diagnostic could be developed for TESSAR. For example,
 a pre-evaluation metric could forecast TESSAR’s sample efficiency gains, helping practitioners decide when to deploy it.

How sensitive is TESSAR to the choice of feature extractor (e.g., vs. raw inputs or other  architectures)? The method relies on a feature mapping (e.g., neural network outputs), and varying this (e.g., with PCA) might affect Voronoi partitions and VLDM scores, warranting empirical sensitivity analysis.

In the theoretical analysis, the Lipschitz assumption is reasonable, but are there empirical cases where it fails, and how does TESSAR perform there?

---

> ### Author Response · Authors · 2025-11-26
>
> We sincerely thank the reviewer for their careful reading, for highlighting multiple strengths of our paper, and for providing several constructive comments. We are particularly pleased with the positive outlook given to our work. Below, we address each concern point-by-point and clarify the aspects raised in the review.
>
> ----
>
> ### W1. Runtime of TESSAR on large datasets
> We acknowledge that TESSAR incurs a higher runtime than the competing baselines. This behavior is expected, as TESSAR must generate multiple perturbed hypotheses and dynamically update VLDM during the acquisition step. Consequently, when the unlabeled pool becomes very large, the runtime gap can grow further. Nevertheless, we note that this scalability issue can be mitigated in practice. In large-scale active learning classification, such as on ImageNet, it is common not to use the entire unlabeled set as the pool. Instead, a randomly sampled subset of the unlabeled data is used as the effective pool before running the acquisition function. The same subsampling strategy can be applied directly to regression, thereby controlling the effective pool size and avoiding prohibitive runtime. Thus, while TESSAR is indeed slower due to its design, this limitation can be circumvented in realistic large-scale scenarios by adopting standard subsampling practices widely used in active learning.
>
> ---
>
> ### W2. Detailed comparison with LCMD
> Thank you for the constructive comment regarding the Related Works section. While both LCMD and TESSAR aim to balance diversity and representativity in regression, the two approaches differ in several fundamental ways. LCMD is a clustering-based method that selects samples by first identifying the largest cluster and then choosing the point farthest from its center. This strategy captures broad representativeness but relies entirely on cluster geometry, potentially leading to the selection of points at the outer periphery of clusters rather than at informative interior boundaries where competing influences from samples intersect. TESSAR also incorporates diversity- and density-oriented components (DIST and BIN) that resemble parts of LCMD’s design. However, the key distinction is the inclusion of VLDM–a perturbation-based geometric measure that explicitly targets samples near Voronoi faces, which correspond to interior regions. Unlike LCMD, which provides no mechanism to prevent the selection of outer points, TESSAR naturally avoids peripheral regions because Voronoi faces do not form along the outer boundary of the data distribution; thus, outer samples cannot exhibit cell-assignment instability. This ensures that TESSAR consistently prioritizes informative interior boundaries and avoids the drift toward uninformative outer regions. This fundamental difference explains why TESSAR demonstrates stronger and more stable performance across datasets. We have incorporated this clarification into the revised Related Works section.
>
> ---
>
> ### Q1. Pre-evaluation of TESSAR compared to random sampling (as in LCDM)?
> Thank you for pointing us in an interesting direction. Following LCMD’s diagnostic approach, we examined whether TESSAR also allows a pre-evaluation of its expected improvement over random sampling. By computing the initial RMSE/MAE ratio and correlating it with TESSAR’s eventual performance gain, we found a strong positive correlation, indicating that this metric can reliably forecast when TESSAR will be beneficial. We have added this analysis in Appendix C.6.
>
> ---
>
> ### Q2. Sensitivity of TESSAR to the choice of feature extractor
> Thank you for the question. We conducted a sensitivity study using raw inputs, gradient-sketching features, GP posterior features, and last-layer embeddings, and found that TESSAR’s performance is largely unchanged across feature maps, with only GP posterior features showing mild degradation. This demonstrates that TESSAR is generally robust to the choice of feature extractor. We have added these results in Appendix C.5.
>
> ---
>
> ### Q3. Regarding the Lipschitz assumption and its impact on TESSAR's performance
> Thank you for this excellent question. First, we could not observe any empirical cases where TESSAR suddenly fails to perform well. While the Lipschitz assumption is well-supported for standard neural network architectures [1,2], the most direct answer to your concern about empirical failure cases lies in our design choice: we compute VLDM solely with respect to the last-layer features (Section 2.3). Since the mapping from these features to the output in a regression task is typically a simple linear layer, the required Lipschitz continuity for this final transformation is inherently satisfied.
>
>
> [1] K. Scaman and A. Virmaux. “Lipschitz regularity of deep neural networks: analysis and efficient estimation.” In NeurIPS 2018.
>
> [2] G. Khromov and S. P. Singh. “Some Fundamental Aspects about Lipschitz Continuity of Neural Networks.” In ICLR 2024.

---

### Official Review · Reviewer_brsN · 2025-10-31

**Soundness:** 3
**Presentation:** 2
**Contribution:** 3
**Rating:** 4
**Confidence:** 4

**Summary:**

The paper discusses TESSAR - an active learning method for regression that picks new points to label by using geometrical structure. TESSAR builds a Voronoi diagram around the currently labeled points and seeks unlabeled samples that lie near the boundaries of the diagram, where the model is least certain. It then balances this with two complementary signals: a score that encodes the density of points or how representative they are, and a score the encodes diversity by measuring distances from labeled data. The result is a single scoring rule that aims to be informative, diverse, and representative. Across many tabular datasets, this approach matches or beats strong baselines, with a practical update trick to keep computation reasonable.

**Strengths:**

The paper presents a clear geometrical idea for the solution of a well motivated problem in active learning, which is integrated into a unified comprehensive strategy.
Attention is given to computational complexity.

**Weaknesses:**

The technical sections are hard to follow, as several derivations are terse.
The experiments are limited to modest-size tabular datasets; given the inherent computational complexity of the method, more challenging datasets seem appropriate.

The authors appear unfamiliar with several active learning works that address a similar problem via coverage. Although those papers target classification at the low-budget regime, their focus - selecting spatially diverse and representative points from the underlying distribution - is closely related. Instead of employing a Voronoi diagram, the optimization is formulated in terms of set coverage; crucially, because the objective is submodular, the greedy solution enjoys efficient approximation guarantees. Published extensions in that line of work also incorporate uncertainty terms. It is therefore essential to compare against this coverage-based literature.

1) Yehuda, Ofer, et al. "Active learning through a covering lens." Advances in Neural Information Processing Systems 35 (2022): 22354-22367.
2) Bae, Wonho, Junhyug Noh, and Danica J. Sutherland. "Generalized coverage for more robust low-budget active learning." European Conference on Computer Vision. Cham: Springer Nature Switzerland, 2024.

**Questions:**

Please address the relationship to the coverage-based literature discussed above, and clarify the method’s scalability to larger datasets.

---

> ### Author Response · Authors · 2025-11-26
>
> We would like to thank the reviewer for highlighting the strengths of our paper (a clear geometric idea for active learning and good discussions of computational complexity) and for providing several constructive comments. Let us clarify the points raised in the review.
>
> ----
>
> ### W1. The technical sections are hard to follow, as several derivations are terse.
> Thank you for your feedback regarding the clarity of the technical sections. We appreciate you pointing this out, and we are committed to improving the readability of our derivations and algorithms.
> To address this, we have revised several parts of the manuscript to ensure the logical flow is easier to follow. Specifically, we have updated the pseudocode as follows:
> - **Algorithm 1:** We now explicitly denote the use of multiple sigmas for the VLDM computation to clarify the procedure.
> - **Algorithms 2 & 3:** We have merged these into a single algorithm to provide a unified view of the main active learning loop. Algorithm 3, previously presented, is now presented as a clearly defined subroutine within the main active learning process.
>
> We hope these changes improve the clarity. If there are other specific derivations or sections you found particularly terse, please let us know, and we will happily expand on them in the next revision.
>
> ----
>
> ### W2 & Q2. The experiments are limited to modest-size tabular datasets; given the inherent computational complexity of the method, more challenging datasets seem appropriate.
> Thank you for pointing out the limitation regarding the dataset scale. We agree that TESSAR has higher computational complexity than other baselines due to its perturbation-based VLDM evaluation and dynamic Voronoi updates. However, similar to large-scale classification active learning, this issue can be mitigated by subsampling the unlabeled pool at each acquisition step rather than operating on the full dataset. This common practice enables practitioners to apply TESSAR to large datasets while keeping the effective pool size manageable. In addition, we have begun running experiments on a substantially larger dataset with roughly 2 million unlabeled samples to further assess TESSAR’s scalability in more challenging settings. Because these runs require considerable time, we are unable to complete the full analysis within the rebuttal period. We will include the results and discussion in the final version of the paper.
>
> ----
>
> ### W3 & Q1. Comparisons with coverage-based active learning
> Thank you for pointing out the coverage-based active learning literature. We agree that these approaches are conceptually related to our geometry-aware selection strategy. Following your suggestion, we incorporated coverage-based methods into the Related Works section. We further conducted additional experiments directly comparing our method against the algorithm proposed in [1] (ProbCov). The results are now included in Figure 5, Table 1 (performance), and Table 3 (runtime). The algorithm in [2] could not be evaluated because the authors did not provide an official full implementation code.
>
> [1] Yehuda, Ofer, et al. "Active learning through a covering lens." NeurIPS 2022.
>
> [2] Bae, Wonho, Junhyug Noh, and Danica J. Sutherland. "Generalized coverage for more robust low-budget active learning." ECCV 2024.

---

### Author Response · Authors · 2025-11-26
**General Response**

We thank the reviewers for their time and for providing constructive comments that have helped us significantly improve our manuscript.

We are especially encouraged by the positive feedback highlighting that our approach addresses the active regression task elegantly via the clear geometric concept of VLDM (brsN, w4R2, 5qY9). Reviewers also noted that TESSAR successfully integrates informativeness, diversity, and representativity into a single score (5qY9), demonstrates notable performance (brsN, w4R2, 5qY9), and is well-written (5qY9).

We have carefully revised our manuscript to address the reviewers’ comments. A summary of the changes follows:

* **VLDM Description:** The empirical VLDM description was corrected to match Algorithm 1, specifically regarding the use of multiple perturbation variances rather than a single noise level.
* **Baselines:** A performance comparison with the covering-lens (ProbCov) baseline was added.
* **Ablation Studies:** We added several ablation studies, including a margin-based TESSAR (Appendix C.2), a static VLDM baseline (Appendix C.3), a feature-map sensitivity study (Appendix C.5), and a pre-evaluation diagnostic following the LCMD paper (Appendix C.6).
* **Scalability:** The discussion on runtime and scalability was expanded to include notes on N-dependence and pool subsampling.
* **Related Works:** This section was expanded to include distribution/coverage-based methods and to clearly highlight the conceptual differences between LCMD and TESSAR.
* **Corrections:** The runtime results table in the original submission contained a one-column shift error; this has been corrected in the revised version.
* **Editorial:** Minor editorial improvements were made, including typo fixes in Sections 2.2 and 2.3.

For clarity, these updates have been highlighted in $\color{red} \textbf{red}$ in our updated manuscript.

We hope that our responses and the revision address all of your concerns. We are happy to answer any further questions the reviewers may have and look forward to engaging in the discussion period.

Thanks,
Authors

---

### Meta-Review · Area_Chair_akVZ · 2026-01-15

**Summary:**

TESSAR proposes a geometry-aware active regression AL method using a Voronoi-based disagreement proxy (VLDM) to target informative interior regions, combined with distance (periphery) and density (representativity). Reviews are broadly positive on motivation, formulation, and consistent empirical gains; remaining concerns were mainly clarity, scaling/runtime, and positioning vs coverage-based AL.

**Reviewer Concerns:**

Most key concerns were addressed in the rebuttal: they corrected the VLDM description, added a direct comparison to ProbCov (covering-lens), expanded related work, added static/dynamic and proxy ablations (margin, etc.), included feature-map sensitivity and a pre-evaluation diagnostic, and clarified multi-scale perturbations plus scalability via pool subsampling. The only outstanding issue is limited evidence on truly large-scale/million-sample settings, but the mitigation discussion is reasonable for a poster.

**Reviewer Scores:**

brsN: likely moves from 4 → 5 (coverage comparison + clarity/scaling additions help).
5qY9: stays 8.
w4R2: stays 8.

---

### Decision · Program_Chairs · 2026-01-26

Accept (Poster)